# Which Surrogate Marker of Insulin Resistance Among Those Proposed in the Literature Better Predicts the Presence of Non-Metastatic Bladder Cancer?

**DOI:** 10.3390/jcm14082636

**Published:** 2025-04-11

**Authors:** Giovanni Tarantino, Ciro Imbimbo, Matteo Ferro, Roberto Bianchi, Roberto La Rocca, Giuseppe Lucarelli, Francesco Lasorsa, Gian Maria Busetto, Marco Finati, Antonio Luigi Pastore, Yazan Al Salhi, Andrea Fuschi, Daniela Terracciano, Gaetano Giampaglia, Roberto Falabella, Biagio Barone, Ferdinando Fusco, Francesco Del Giudice, Felice Crocetto

**Affiliations:** 1Department of Clinical Medicine and Surgery, University of Naples “Federico II”, 80131 Naples, Italy; 2Department of Neurosciences, Reproductive Sciences and Odontostomatology, University of Naples “Federico II”, 80131 Naples, Italy; ciro.imbimbo@unina.it (C.I.); robertolarocca87@gmail.com (R.L.R.); gaetanogiampaglia@hotmail.it (G.G.); felice.crocetto@unina.it (F.C.); 3Unit of Urology, Department of Health Science, University of Milan, ASST Santi Paolo e Carlo, 20172 Milan, Italy; matteo.ferro@unimi.it (M.F.); roberto.bianchi.urol@gmail.com (R.B.); 4Urology, Andrology and Kidney Transplantation Unit, Department of Emergency and Organ Transplantation, University of Bari, 70124 Bari, Italyfrancesco-lasorsa96@libero.it (F.L.); 5Department of Urology and Renal Transplantation, University of Foggia, 71122 Foggia, Italy; gianmaria.busetto@unifg.it (G.M.B.); marco.finati@gmail.com (M.F.); 6Department of Urology, Sapienza University of Rome, 04100 Latina, Italy; antonioluigi.pastore@uniroma1.it (A.L.P.); yazan.alsalhi@uniroma1.it (Y.A.S.); andrea.fuschi@uniroma1.it (A.F.); 7Department of Translational Medical Sciences, University of Naples “Federico II”, 80131 Naples, Italy; daniela.terracciano@unina.it; 8Urology Unit, Azienda Ospedaliera San Carlo, 85100 Potenza, Italy; rfalabella@libero.it; 9Department of Urology, Ospedale San Paolo, ASL NA1 Centro, 80147 Naples, Italy; biagio.barone@aslnapoli1centro.it; 10Division of Urology, Department of Surgical Sciences, AORN Sant’Anna e San Sebastiano, 81100 Caserta, Italy; ferdinando-fusco@libero.it; 11Department of Urology, University Sapienza, 00185 Rome, Italy; francesco.delgiudice@uniroma1.it

**Keywords:** bladder cancer, insulin resistance, surrogate markers of insulin resistance, triglyceride–glucose index, AUROC

## Abstract

**Background**: Recent evidence has shown that insulin resistance (IR), a hallmark of nonalcoholic fatty liver disease, predicts bladder cancer (BC) presence. However, the best surrogate marker of IR in predicting BC is still unclear. This study examined the relationships among ten surrogate markers of IR and the presence of BC. **Methods**: Data from 209 patients admitted to two urology departments from September 2021 to October 2024 were retrospectively analyzed. Individuals (median age 70 years) were divided into two groups (123 and 86 patients, respectively) based on the presence/absence after cystoscopy/TURB of non-metastatic BC. Univariate logistic regression was used to determine the relationships between groups, and the following IR parameters: Triglyceride–Glucose (TyG) index, TyG-BMI, HOMA-IR HOMAB, MetS-IR, Single Point Insulin Sensitivity Estimator, Disposition Index, non-HDL/HDL, TG/HDL-C ratio and Lipoprotein Combine Index. Stepwise logistic regressions were carried out to evaluate the significant predictions and LASSO regression to confirm any significant variable(s). The predictive value of the index test for coexistent BC was evaluated using receiver operating characteristic (ROC) curves and the area under the ROC curve (AUC). **Results**: The univariate analysis revealed that the TyG index and MetS-IR were associated with the BC presence. Specifically, the associations of the TyG index and MetS-IR were more significant in participants =/> 65 years old. In multivariate analysis, the stepwise logistic regression, evaluating the most representative variables at univariate analysis, revealed a prediction of BC by only TyG index (OR 2.51, *p* = 0.012), confirmed by LASSO regression, with an OR of 3.13, *p* = 0.004). Assessing the diagnostic reliability of TyG, it showed an interesting predictive value for the existence of BC (AUC = 0.60; 95% CI, 0.51–0.68, cut-off 8.50). Additionally, a restricted cubic spline model to fit the dose–response relationship between the values of the index text (TyG) and the BC evidenced the presence of a non-linear association, with a high predictive value of the first knot, corresponding to its 10th percentile. The decision curve analysis confirmed that the model (TyG) has utility in supporting clinical decisions. **Conclusions**: Compared to other surrogate markers of IR, the TyG index is effective in identifying individuals at risk for BC. A TyG threshold of 8.5 was highly sensitive for detecting BC subjects and may be suitable as an auxiliary diagnostic criterion for BC in adults, mainly if less than 65 years old.

## 1. Introduction

Bladder cancer (BC) is the 9th most frequent cancer worldwide, with more than 614,298 new cases of BC occurring in 2022 [1]. The primary factor causing BC, even in industrialized societies, is cigarette smoking. But, specific chemicals, deriving from a variety of occupational exposures, have also been identified [2]. Moreover, metabolic syndrome (MetS) and its components are associated with increased susceptibility and poor prognosis of BC [3,4,5,6]. MetS and its components–dyslipidemia, hypertension, type 2 diabetes mellitus (T2DM), and obesity–share a common mechanism, insulin resistance (IR), which is an almost universal finding in nonalcoholic fatty liver disease (NAFLD, [7]). Interestingly, previous clinical and epidemiological lines of evidence proved that IR is a risk factor for various types of tumors [8]. Very recent findings from a systematic review and meta-analysis confirm that patients with a cancer diagnosis are markedly insulin-resistant [9]. Accordingly, in a previous study, a higher prevalence of IR was found in non-metastatic BC patients [10]. IR is canonically investigated by the hyperinsulinemic–euglycemic clamp technique [11]. However, feasibility matters of the glucose clamp procedure prevent its use in epidemiological studies. Therefore, a number of simple surrogate indices of insulin resistance and sensitivity have been developed, validated, and, in everyday practice, utilized, such as the Homeostasis Model Assessment (HOMA-IR), [12], HOMA-β cell [13], Triglyceride–Glucose index (TyG, [14]), TyG-BMI [15], Triglycerides/HDL-Cholesterol ratio (TG/HDL-C ratio, [16]), Disposition Index (DI, [17]), Metabolic Score for Insulin Resistance (MetS-IR, [18]), Single Point Insulin Sensitivity Estimator (SPISE, [19]), Non-HDL-cholesterol/HDL-cholesterol ratio (NHHR, [20]), and Lipoprotein Combine Index (LCI, [21]).

As metabolic dysfunction is correlated not only with an increased risk of BC but also with higher recurrence rates and reduced overall survival, the need to find a test mirroring IR, useful to screen for the presence of BC, is compelling. Consequently, the aim of the present study was that of comparatively studying the performance of several surrogate markers of IR in order to establish which one could be more reliable in helping ascertain which patients likely suffer from this cancer, for the purpose of more effectively and selectively managing them.

## 2. Methods

### 2.1. Study Design

The study at hand was a retrospective analysis of clinical and laboratory data collected from records of 238 consecutive patients, as they were admitted to two departments of a tertiary university/hospital for symptoms/findings of urinary tract diseases undergoing cystoscopy or TURB, from September 2021 to October 2024. The whole population was split into two distinct groups, i.e., patients with histologically confirmed, non-metastatic BC n 139 and patients suffering from non-cancerous bladder diseases (No BC) n 99. Furthermore, 16 patients of the BC group and 11 from the group without bladder cancer were not taken into account due to the co-presence of diseases. Fourteen patients suffering from liver cirrhosis of any etiology were excluded due to the potential impact of this disease on cholesterol levels, as well as eight patients on anti-lipid therapy, and, finally, five subjects with alcohol abuse, according to [22]. In total, 209 (123 with BC and 86 with No BC) participants were included in this study. The control group (patients without BC) was formed from 35 males suffering from benign prostatic hyperplasia, seven from overactive bladder (OB), 12 from hematuria of unknown origin, 22 from bladder stones, and seven females suffering from recurrent urinary tract infections (UTIs) and three from OB. Retrospective designs provide a vehicle for research using existing data but can be riddled with threats to both internal and external validity. To minimize such treads, a control group was set up [23].

### 2.2. Anthropometric Evaluation

Normal weight was considered as body mass index (BMI) between 18.5 and 24.9, overweight a BMI between 25 and 29.9, and obesity was characterized by a BMI of 30 or more.

### 2.3. Clinical Assessment

Type 2 diabetes mellitus (T2DM) was diagnosed in the presence of fasting plasma glucose (FPG) concentrations ≥ 126 mg/dL, or a glycated hemoglobin test = 6.5 or higher, or on anti-diabetic agents. Prediabetes was assessed by FPG levels falling between 100 mg/dL and 125 mg/dL or a glycated hemoglobin test between 5.7% and 6.4%. Hypertension was diagnosed when the blood pressure reading was equal to or greater than 130/80 mm Hg. Data for systolic/diastolic blood pressure were the average of three consecutive detections taken after allowing the subjects to rest for five minutes in the sitting position. Subjects on current anti-hypertensive drugs were considered as suffering from hypertension, even if it was controlled. Patients on anti-diabetic agents maintained their schedule for metabolic compensation. Participants were categorized into three groups based on their smoking history: non-smokers, former smokers, and current smokers. Being the chronological age greater than or equal to 65 years commonly accepted as the definition of elderly, the patients, in a collateral analysis, were divided into two groups based on this cut-off, according to a previous study [24].

Chronic kidney failure was ascertained following the National Kidney Foundation criteria [25].

### 2.4. Diagnostic Criteria of Bladder Cancer

Cystoscopy was followed by Biopsy/Transurethral Resection of Bladder Tumor (TURBT). The grade and the stage of BC were established by examining the tissue sample, removed during a TURB procedure from the area where cancer may have existed, following the recent well-accepted guidelines [26].

The ethics committee of Federico II University Medical School of Naples, Italy, granted approval to the study (n° 235/2020, dated 14 July 2020), and all individuals gave their informed consent.

### 2.5. Surrogate Markers of Insulin Resistance and Their Calculation

NHHR = Total cholesterol (mg/dL)-HDL/HDL-Cholesterol (mg/dL); TyG index =  Ln [Triglycerides (mg/dL) * FPG (mg/dL)/2]; TyG-BMI =  TyG index * BMI; TG/HDL-C ratio  =  Triglycerides (mg/dL)/HDL-Cholesterol (mg/dL) ratio; HOMA-IR = FPG (mg/dl) * Insulin mIU/L/405; HOMA-B (β cell) = 20 * fasting insulin (mU/L)/[FPG (mmol/L) − 3.5]; Disposition index (DI): HOMA-B/HOMA-IR ratio; MetS-I  =  Ln [(2 * FPG (mg/dL))  +  TG (mg/dL)] * BMI (kg/m^2^)/(Ln [HDL-Cholesterol(mg/dL)]); SPISE = 600* HDL-Cholesterol ^0.185/(Triglycerides ^0.2 * BMI^1.338); LCI = total Cholesterol * Triglyceride * Low-Density Lipoprotein.

The lipid profile–indicating the presence of dyslipidemia–comprehended serum levels of triglycerides (n.v. < 150 mg/dL), total cholesterol (n.v. < 200 mg/dL), HDL-Cholesterol (n.v. > 40 mg/dL for men and >50 mg/dL for women), and low-density lipoprotein (n.v. < 100 mg/dL), which were measured according to in-house procedures.

### 2.6. Statistics

Variables not normally distributed according to the Shapiro–Wilk test analysis were expressed as a median plus interquartile range, while those derived from a normally distributed population were reported as mean plus standard deviation. For every examined variable, the number of observations was pointed out to interpret results accurately. Differences between medians of the groups were detected by the two-sample Wilcoxon rank-sum (Mann–Whitney) test, while the difference between means was analyzed by the independent t-test. The extended Mantel–Haenszel with ANOVA (transformation in ranks) analysis [27] was used when evaluating differences between the two groups adjusted for gender. The two-way table with measures of association and the related Pearson’s chi-squared test were used to weigh frequencies when dealing with categorical variables.

Among particular methods of regression analysis used for prediction, an ordered logistic model was chosen to evaluate the relationship between a binary dependent variable and an ordinal independent variable, such as BMI classes and smoking habit. Vice versa, an ordered probit model (robust, if dealing with outliners) was applied to estimate relationships between an ordinal dependent variable, i.e., grading or staging, and an independent variable (such as surrogate markers of IR, categorized in four cuts, i.e., three quartiles). In the univariate analysis, the logistic regression was carried out to predict the presence/absence of BC, evaluating the odds ratio and related parameters. The *Z* scores were used for the normalization of each parameter, facilitating the comparison of OR for each standard deviation increase. Pseudo R-square, as a statistical measure of whether the model better predicts the outcome, was also reported. The Hosmer–Lemeshow goodness-of-fit test was used to evaluate the calibration of the prediction model, with a non-significant test indicating a good fit. To further ascertain whether independent variables are ‘significant’ for a model or not, the Wald statistic was carried out, running a generalized estimating equation (significance if < 0.05), according to https://www.stata.com/manuals/xtxtgee.pdf (accessed on 12 January 2025). In the multivariate analysis, a stepwise method (*p* for enter <0.10) helped us simplify our model, reduce overfitting, and improve prediction accuracy. Specifically, among the univariate-selected variables, we studied which one(s) remained in the model. A conservative threshold of the VIF value (<5) was selected to indicate no significant multicollinearity issues among the variables. The stepwise method, being sensitive to sample size, as well as to the order of variables and the correlation among variables, was supplemented by the Double Selection LASSO (Least Absolute Shrinkage and Selection Operator) Login Regression (DSLLR) plug-in formula. DSLLR offers a neat way to model the dependent variable while automagically selecting significant variables by shrinking the coefficients of unimportant predictors to zero when the number of covariates is small and fixed.

To screen the reliability of the diagnostic test, its sensitivity and specificity were assessed. Furthermore, the likelihood ratios (LR) summarized the diagnostic accuracy using the menu *estat classification* after the logistic regression. A LR+ greater than 1 means that a positive test is good at ruling in a diagnosis. On the contrary, a LR lower than 1 means that a negative test is good at ruling out a diagnosis. The cut-off with the highest specificity and sensitivity was calculated by means of the Youden Index according to [28]. The decision curve analysis after the logistic regression was performed to help address the clinical utility of using a surrogate marker to predict the BC presence, according to [29]. The Lowess, which is a simple and powerful strategy for fitting smooth curves to empirical data, was applied. Consequently, a restricted cubic spline model (three knots) to fit the dose–response relationship between the index test(s) and the BC was carried out, rejecting the null hypothesis that the odds of BC is a linear function of mean TyG when *p* > 0.05. Finally, a segmented logistic regression was performed. Still, the ROC analysis (DeLong method) was used in diagnostic decision-making of the BC presence/absence. To measure the performance of the binary classification test (index test), the area under the receiver operating characteristic curve (AUROC/AUC) was evaluated to identify the most appropriate models (the highest specificity and sensitivity) under the nonparametric assumption, i.e., distribution free. As a post-estimation, the sensitivity/specificity versus probability cut-off plot indicated where the intersection point between sensitivity and specificity lies, in order to show the accuracy of the index test. As is common in literature, the authors chose a 50% cut-off to predict the positive and negative values. This cut-off is useful for trying to balance the harms of the type I and type II errors. As a post-hoc exploratory examination, the power analysis to establish the minimum sample size was performed by calculating the difference of means and SDs (two-sided) of the main outcome in the two groups (No BC and BC patients). A *p*-value < 0.05 was accepted as significant. Statistical analyses were run on StataNow 18.5., Stat Corp., College Station, TX, USA.

## 3. Results

The main demographic features, co-morbidities, and the metabolic assets of the two groups are shown in Table 1.

The two cohorts were well-matched for age, gender and BMI (Pearson’s chi^2^; *p* = 0.56, 0.17 and 0.37, respectively. Only the percentages of the smoking habit and the serum values of triglycerides were statistically different between the examined groups, *p* = 0.007 and 0.02, respectively. The frequencies of other co-morbidities, such as T2DM, hypertension, dyslipidemia and kidney failure, were similar in both the subsets.

Among the surrogate markers taken into account, i.e., HOMA-IR, HOMA-B, DI, MetS-IR, SPISE, TyG, TyG-BMI, TG/HDL ratio and non-HDL/HDL ratio, only MetS-IR and TyG showed a significant difference (*p* = 0.03 and = 0.02, respectively), as is evident in Table 2. As a collateral finding, the TG/HDL ratio had a borderline significance.

Between the examined groups, there was a significant difference concerning the values of TyG index and Met-S, two-sample Wilcoxon rank-sum (Mann–Whitney) test, *p* = 0.02 and *t*-test, *p* = 0.03), respectively. When TyG and MetS-IR values were adjusted for gender, the previous statistical difference remained (Extended Mantel–Haenszel, *p* = 0.02 and 0.03, respectively), showing that gender is not a confounding factor. When adjusted for age by the same statistical tool, TyG maintained the significance (*p* = 0.03), while it was lost for MetS-IR (*p* = 0.09). The frequency of dyslipidemia tended to be greater in the BC group. There was no difference in the median values in the remaining markers of insulin sensitivity/resistance.

The post-hoc power analysis established that the total minimum sample was 238 patients with 119 for each group showing a power of 80% and an alpha value of 0.05 on the basis of the TyG levels, showing a modestly reduced sample size of the control group.

## 4. Predictions

In the ordered logistic regression, neither the smoking habit (*p* = 0.51), nor the T2DM (*p* = 0.20), nor the hypertension (0.47), nor the kidney failure presence (*p* = 0.40) predicted the diagnosis of BC. Again, dyslipidemia showed an interesting trend in this regard (*p* = 0.063). By the same statistical tool, the three classes of BMI did not predict the BC presence (*p* = 0.23), while the 1st and the 3rd quartile of TyG significantly predicted the BC grade and stage (Coefficient. 0.16 Robust Std. err.: 0.8, z: 2.05, *p* > |z|: 0.04, 95% conf. interval: 0.007–0.31 and Coefficient: 0.17 Robust Std. err.: 0.007, z: 2.31 *p* > |z|: 0.02, 95% conf. interval: 0.026–0.42, respectively). The characteristics of the grade and stage of BC patients are shown in the Appendix A. The odds ratios of the ten surrogate markers in predicting the BC presence at univariate analysis were evidenced in Table 3.

With a note of caution (low pseudo R^2^) in their interpretation, the only two significant markers were TyG and MetS-IR. Both the goodness-of-fit tests after the logistic model for TyG and MetS-IR showed a high *p*-value, i.e., 0.42 and 0.45, respectively. The Wald test turned out to be significant, i.e., for TyG *p* = 0.02 and for MetS-IR *p* = 0.03, according to generalized estimating equations, meaning that including these variables created a statistically significant improvement in the fit of the model.

Dealing with the sensitivity analyses (Table 4), in the age-stratified analysis, it became clear that the increase in TyG and in MetS-IR is more predictive of the risk of displaying BC among less old individuals compared to older adults.

In the multivariate analysis, among the variables selected by the univariate analysis, i.e., HOMA-IR, TG/HDL-C ratio, MetS-IR, TyG-BMI and TyG index, only the TyG index remained in the model (odds ratio 2.51, Std. err. 0.012, z 2.51, *p* > |z| 0.012, 95% conf. interval. 1.25–6.11, stepwise logistic regression, *p* for enter (<0.10). The VIF value for multicollinearity was very low, i.e., 1. The lasso logistic regression (DSLLR) evaluating the relationship between BC presence and all surrogate makers of IR, such as TyG, TyG-BMI, HOMA-IR, HOMA-B, MetS-IR, SPISE, non-HDL/HDL, TG/HDL-C ratio, DI, and LCI controlled for age, gender, BMI and smoking habit, retained actively only TYG, with an odds ratio of 2.73, *p* > |z| 0.006, 95% CI 1.36–6.61.

The diagnostic accuracy of TyG was characterized at the logistic regression: Sensitivity, 75.76%; Specificity, 36.59%; Positive predictive value, 59.06%; Negative predictive value, 56%; False positive rate for true no BC, 63.41%; False negative rate for true BC, 24.24; False positive rate for classified positive no BC, 40.94%; False negative rate for classified negative BC, 44.44%; Correctly classified, 58.01; Positive likelihood ratio, 3.12; Negative likelihood ratio, 1.19; cut-off calculated as Youden index, 8.50.

The LR+ of TyG indicates that a patient with BC is 3.12 times as likely to test positive as someone without the disease. Vice versa, the LR-values of the same test mean that a patient with BC is only 0.19 times as likely to test negative as someone without the disease. The classification is shown in Table 5.

The Hosmer–Lemeshow goodness-of-fit test was used to evaluate the calibration of the prediction model, and the results indicated x^2^ = 7.37, *p* = 0.49, suggesting no statistically significant difference between the predicted value of the model and the actual observed value. The calibration curve showed that the nomogram had good predictive stability and consistency with an overlapping significance. As shown in Figure 1, the predictive model exhibited good calibration ability.

The decision curve analysis (DCA) helped us address the clinical utility of using increased levels of TyG (cut-off 8.5) to predict BC presence and eventually treat it, as shown in Figure 2.

After employing the Lowess, the authors rejected the null hypothesis that the odds of BC is a linear function of mean TyG, as shown in Figure 3.

Additionally, employing a restricted cubic spline (RCS) model (three knots) to fit the dose–response relationship between TyG values and the BC, the presence of non-linear associations was further assessed, Prob > chi2 = 0.021. Consequently, a segmented logistic regression (knot 1 = 7.9, knot 2 = 8.5, knot 3 = 9.1, corresponding to the 10th, 50th. and 90th percentile of the TyG values) was performed. It showed that only the first knock, i.e., the first quartile, had a significance with odds ratio: 2.1; Std. err.: 0.679; z: 2.30, *p* > |z|: 0.022; 95% CI: 1.11–3.95. This finding further validated the results of the previous association analysis.

Finally, when ascertaining the BC presence, the predictive ability/diagnostic reliability of the TYG levels was confirmed by the AUROC, being nearly 0.60, 95% CI: 0.51–0.68, as evidenced by Figure 4.

Furthermore, a plot of sensitivity/specificity versus probability cut-off point was assessed, showing an adequate discrimination power, as observed in Figure 5.

## 5. Discussion

This study investigated the predictive accuracy and discriminative abilities of ten surrogate markers of IR in diagnosing which patients could suffer from BC. Earlier studies have shown that IR, resulting from increased visceral adiposity [30], plays a central role in the increased mortality of patients with cancer contextually affected by obesity [31], and, in particular, that elevated BMI is associated with BC recurrence and progression [32]. Indeed, those previous pieces of research have not explicitly addressed which laboratory text, mirroring IR, is the most reliable in assessing the BC presence. In this context, a very recent paper has dealt with the presence of NAFLD in BC patients [10], using TyG as a single surrogate marker of IR, without confronting it to the well-accepted others. The results of the present study showed that TyG, differently from other tests assessing IR, is effective in identifying individuals at risk for BC. A TyG threshold of 8.5 was highly sensitive for detecting BC subjects and may be suitable as an auxiliary diagnostic criterion for BC in adults, mainly in those less than 65 years old. Interestingly, TyG quartiles also predicted the BC grading and staging.

Generally, surrogate markers of IR were, and still are, utilized to identify MetS and its components [33], to assess the progression of atherosclerotic-related conditions including hypertension, inflammation, endothelial dysfunction, and coronary artery disease [34], to diagnose chronic kidney diseases [35], or for evaluating the cognitive decline [36].

Recently, some authors found that the TyG index, TG/HDL-C ratio, and MetS-IR were positively correlated with colorectal cancer incidence in Koreans [37]. Similarly, elevated values of the TyG index, TG/HDL-C ratio and MetS-IR were strongly associated with the occurrence of young-onset colorectal adenomas in individuals under 50 years old [38]. The TyG index, TyG-BMI, TG/HDL-C, and MetS-IR were closely associated with the risk of esophageal adenocarcinoma and esophageal squamous cell carcinoma [39]. A meta-analysis comprehending 30 studies, encompassing 2,058,536 participants, showed that elevated TyG index is significantly linked to a heightened risk of breast cancer [40]. Among the lipid accumulation product (LAP), the TyG index, the TyG-BMI, and the high quartiles of the TyG index and LAP were related to gastrointestinal cancers in the US population [41]. The TyG index was significantly correlated with non-small cell lung cancer risk, [42].

Existing studies showed that IR indirectly stimulates tumor growth through NF-κB, which supplies a mechanistic link between inflammation and cancer, and is a central factor controlling the ability of both pre-neoplastic and malignant cells to oppose apoptosis-based tumor-surveillance molecular events [43]. The inflammatory tumor microenvironment created by chronic inflammation can prompt ROS production through inflammatory cells [44]. ROS overproduction, by triggering the P13K/Akt signaling, could induce adverse genetic modifications and DNA damage followed by tumor formation and progression [45]. Advanced glycation end products (AGEs) are made by metabolically active cells, although they start accumulating in the cell under oxidative stress [46].

The production of intracellular AGE precursors damages target cells by modifying proteins and altering their function. The signal transduction receptor for AGEs (RAGE), by activating NF*κ*B, can lead to adverse changes in gene expression [47].

IR increases the fraction of circulating free IGF-I by down-regulating hepatic synthesis of IGFBP-1 [48]. Circulating concentrations of IGF-I are associated with an increased risk of common cancers [49]. The IGF system in cancer should be examined in the context of the extra-cellular and intra-cellular signaling networks, particularly phosphatidylinositol 3-kinase (PI3K), protein kinase B (Akt/PKB), mammalian target of rapamycin (mTOR), and forkhead transcription factors (FOXO) [50].

The risk of BC was associated with urinary tract infections (UTIs) in a unique study [51]. Both IR and diabetes mellitus have shown an increased UTI susceptibility due to the production of adipose-derived circulating cytokines that dysregulate immune mechanisms like NF-κB signaling and the integrated stress response [52]. Recent evidence suggests that the human genitourinary microbiome, the so-called urobiome, plays a significant role in mediating the development and progression of urological tumors, including BC [53]. Specifically, investigators concluded that *E. coli* infection may participate in the development of BC via activation of the NF-κB pathway, inhibiting apoptosis and augmenting inflammation [54]. Indeed, results have also proved that STAT-1 plays a critical role in the regulation of E. coli invasion and infection in the uroepithelial cells, especially those pretreated with glucose [55]. An increased abundance of Porphyromonas and Porphyromonas *somerae* was found in BC patients compared with controls [56]. Confronting the microbiome of BC patients vs. healthy controls, beta-diversity was significantly dissimilar, with *Actinomyces*, *Achromobacter*, *Brevibacterium*, and *Brucella* significantly more abundant in the urine samples of BC patients [57]. Accordingly, there were significant differences in beta diversity in diabetics, i.e., a significantly reduced abundance of butyrate-producing genera was associated with higher HOMA-IR [58]. As an intriguing consideration, as some surrogate markers of IR taken into account also indicate the presence of NAFLD, which was recently renamed “metabolic associated steatotic liver disease”, the link of BC with this very common liver disease should be further deepened in larger scale studies. It is noteworthy to stress that, having enrolled ten patients with OB and likely overactive detrusor, a recent study showed that intradetrusor injections of botulinum toxin in such patients showed no effect on bladder wall inflammation and actually improved the inflammatory condition of the muscle in a significant number of samples [59].

The strength of the present clinical study is that authors have evaluated the performance of multiple surrogate markers of IR, as an add-on diagnostic test. To our best knowledge, ours was one of the few studies addressing surrogate markers of IR in BC patients, even though T2DM and excessive body weight, both closely linked to IR, as components of MetS, are associated with increased susceptibility of BC in a very large population [60]. Regarding limitations of the study, due to its retrospective nature, the authors include the fact that the potential impact of fluctuations in these indicators on their association with BC could not be assessed. Unfortunately, we were not able to eliminate the systematic difference between the ability of participant groups (three) to accurately recall their smoking habit, the main risk factor for BC. According to the WHO, one can be considered a non-smoker five years after he/she quits smoking. https://apps.who.int/iris/bitstream/handle/10665/112833/9789241506939_eng.pdf (accessed on 12 January 2025). On the other hand, we do not know if this definition applies to the risk for BC as well. Moreover, a former smoker might be a person who decided to stop after the first episode of hematuria, and shortly after was diagnosed with BC.

Still, controls may not necessarily be representative of the general population and there could be a risk of bias from resources. Furthermore, the LAP and the estimated glucose disposal rate, both recent markers assessing lipid-based IR and insulin sensitivity [41,61] were not assessed due to the scarcity of reported values of the waist circumference in the analyzed records. Regardless, a recent study showed a surprising overlapping OR of both TyG and LAP in predicting gastrointestinal cancer [62].

## 6. Conclusions

As metabolic dysfunction in patients with BC is associated with increased recurrence and reduced overall survival, future studies should address if assessing its presence early and trying to ameliorate IR, and the linked syndromes, could improve patient care in this population.

Among the surrogate markers of IR taken into account in this retrospective study, the TyG index is reliable in identifying individuals at risk for BC. A TyG threshold of 8.50 was highly sensitive for detecting BC subjects and may be suitable as an auxiliary diagnostic criterion for BC in adults, mainly if they are less than 65 years old.

## Figures and Tables

**Figure 1 jcm-14-02636-f001:**
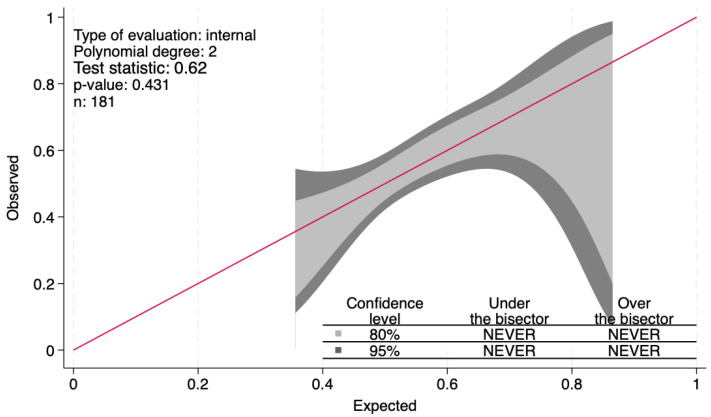
The calibration curve of the triglyceride–glucose index. Legend: The *p*-value of this curve, i.e., 0.43, was similar to that resulting from the Hosmer–Lemeshow goodness-of-fit test, i.e., 0.49, confirming the good calibration ability of the index test. The red line represents the best-fit linear line that shows the correlation between actual and predicted values.

**Figure 2 jcm-14-02636-f002:**
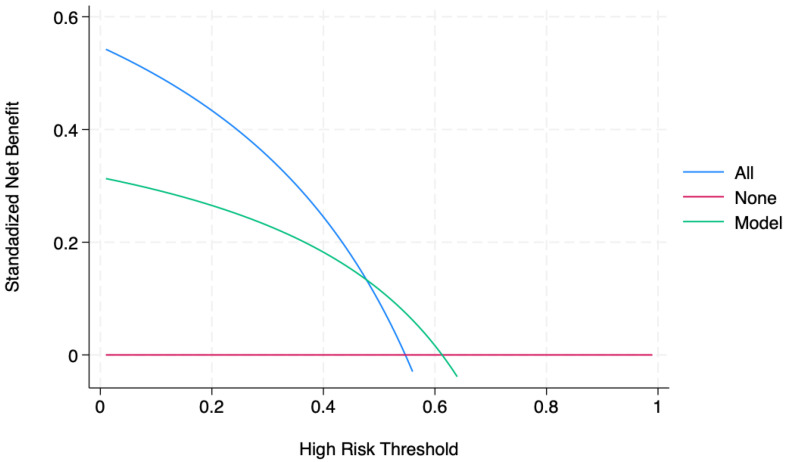
Decision curve analysis. Legend: The clinical decision curve analysis revealed a net positive benefit in a relatively small segment of the population, suggesting a possible clinical application value. Readers can see here that the increased TyG levels add good diagnostic value to a range of threshold probabilities near 57.5–60.5%. For example, if the patient’s high-risk threshold is 60%, then TyG concentration alone can be beneficial in ascertaining the bladder cancer presence. However, if the patient’s threshold probability is less than 57.5% or higher than 60.5%, then using TyG levels to confirm the diagnosis has less benefit than choosing to confirm or not confirm, respectively.

**Figure 3 jcm-14-02636-f003:**
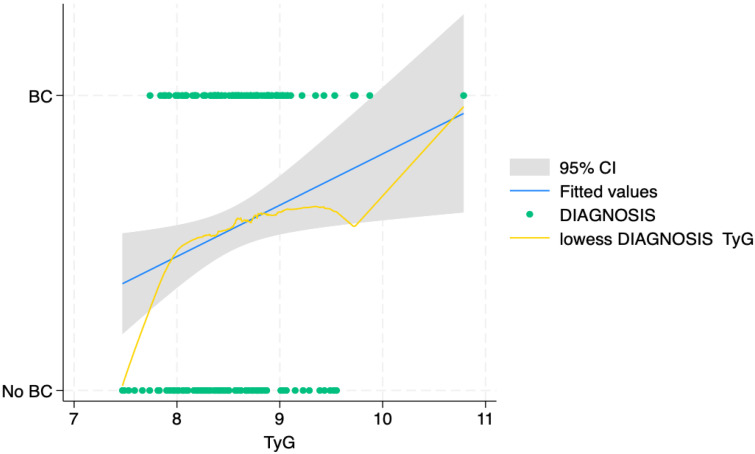
The dose–response relationship curve of triglyceride–glucose index and bladder cancer. Legend: The graphical representation delineated an ascending trend in bladder cancer risk with increasing values TyG, which stabilized past an index value of 7.9 followed by a gradual deceleration. Notably, at the juncture where TyG reached 9.6, there was a resurgence in the increasing trend, p for nonlinearity = 0.0217. Abbreviations: BC, presence of bladder cancer; No BC, absence of bladder cancer.

**Figure 4 jcm-14-02636-f004:**
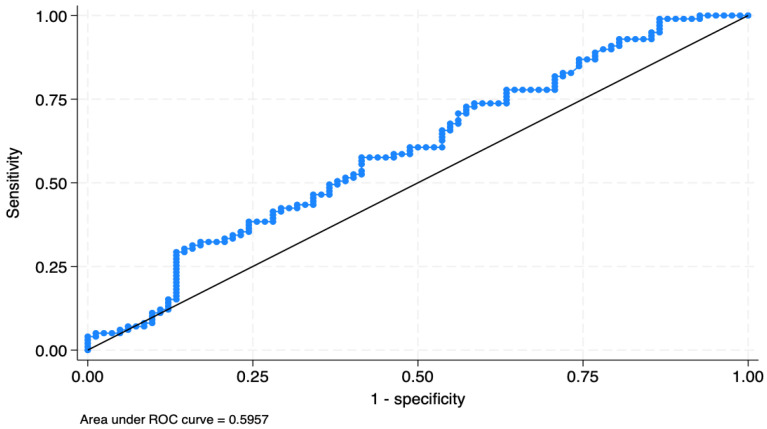
The AUROC of the triglyceride–glucose index. Legend: Receiver Operating Characteristic (ROC) curves for triglyceride–glucose index in predictive analysis. The continuous line represents the line of no discrimination, which corresponds to an AUC of 0.5.

**Figure 5 jcm-14-02636-f005:**
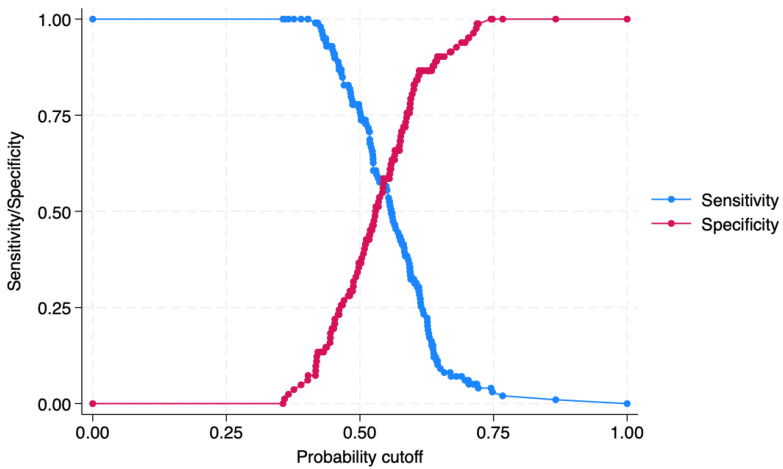
Dealing with post-estimation, the graph of sensitivity/specificity versus probability cut-off plot produced evidence that the sensitivity and specificity for the Triglycerides-Glucose Index is satisfactory, at a probability cut-off of 0.50. In fact, the two lines (red and blue) merged at a level superior to 50%, indicating a sufficient reliability.

**Table 1 jcm-14-02636-t001:** Characteristics of the patients belonging to the cohorts of bladder cancer and non-bladder cancer.

Patients (n)	BC (n = 123)	No BC (n = 86)	*p*
Age years (median/IQR)	70 (62–76)	70 (61/77)	0.56 *
Gender M/F (n)	100/76	76/10	0.17 ^
BMI	26.4 (24.1–29)	25.9 (24–28.7)	0.37 *
BMI 18.5–24.9/25–29.9 /> 30 (n)	39/59,721	33/43/10	0.44 ^
**Smoking yes/no/FMs (n)**	**59/15/48**	**23/17/46**	**0.007 ^**
T2DM yes/no/prediabetes (n)	32/52/36	18/46/22	0.25 ^
Hypertension yes/no (n)	74/49	56/30	0.46 ^
Kidney failure yes/no (n)	9/197	3/83	0.24 ^
Dyslipidemia yes/no (n)	86/37	70/16	0.061 ^
Total cholesterol mg/dL (median/IQR)	180 (153–206)	175 (147–203)	0.55 *
HDL-cholesterol mg/dL (median/IQR)	46 (41–54)	47.5 (41–57)	0.6 *
LDL-cholesterol mg/dL (median/IQR)	117 (91.5–142)	113 (88–139)	0.57 *
**Triglycerides mg/dL** **(median/IQR)**	**110 (80–144)**	**96 (70–128)**	**0.02 ***
FPG mg/dL (median/IQR)	95 (88–109)	95.5 (85–110)	0.62 *
Insulin mIU/L (median/IQR)	9.4 (5.9–14.2)	9.3 (6.9–11.7)	0.37 *
Uric acid mg/dL (median/IQR)	5.8 (4.9–6.8)	5.85 (5.2–6.5)	0.73 *
Creatinine mg/dL (median/IQR)	0.96 (0.82–1.19)	0.96 (0.85–1.12)	0.9 *

Legend: The smoking habit and the triglyceride levels were significantly different between the groups. * Two-sample Wilcoxon rank-sum (Mann–Whitney) test, ^ Pearson’s chi-squared. Abbreviations: n, number of patients; M/F, Male/Female; IQR, Interquartile Range; BMI, Body Mass Index; FMs, Former Smokers; T2DM, Diabetes Mellitus Type 2; HDL, High-Density Lipoprotein; LDL, Low-Density Lipoprotein; FPG, Fasting Plasma Glucose.

**Table 2 jcm-14-02636-t002:** Values of the surrogate markers of insulin resistance in the whole cohort and in the sub-groups.

Surrogate Markers IR Patients (n)	Total Population (n = 209)	BC Group (n = 123)	No BC Group (n = 86)	*p* *
HOMA-IR median (IQR)	2.27 (1.53–3.18)	2.34 (1.48–3.67)	2.18 (1.57–2.85	0.17 *
HOMA-B median (IQR)	1.98 (1.32–2.79)	1.38 (0.91–2.94)	1.88 (1.32–2.77)	0.41 *
DI median (IQR)	0.99 (0.85–1.16)	1.01 (0.85–1.15)	0.99(0.84–1.25)	0.40 *
**MetS-IR** **Mean +/− SD**	**1.04 +/− 0.29**	**1.71 +/− 0.7 °**	**1.74 +/− 0.8 °**	**0.03 °**
SPISE median (IQR)	6.15 (5.36–7.21)	5.88 (5.24–7.21)	6.45 (5.58–7.27)	0.08 *
**TyG median (IQR)**	**8.5 (8.18–8.83)**	**8.58 (8.27–8.89)**	**8.42 (8.05–8.74)**	**0.02 ***
TyG-BMI median (IQR)	223 (201–243)	228 (202–249)	219 (201–235)	0.08 *
TG/HDL-C ratio median (IQR)	2.22 (1.43–2.95)	2.35 (1.59–3.21)	1.96 (1.36–2.85)	0.05 *
non-HDL/HDL ratio median (IQR)	2.65 (2.17–3.38)	2.8 (2.26–3.38)	2–52 (1.09–3.45)	0.2 *
LCI median (IQR)	40,283 (23,864–74,315)	47,051 (26,795–80,527)	35,925 (21,249–73,155)	0.18 *

Legend: TyG and MetS-IR values were significantly different in the two groups. * Two-sample Wilcoxon rank-sum (Mann–Whitney) test, ° Two-sample t-test. Abbreviations: BC, Bladder Cancer; BMI, Body Mass Index; TyG, Triglyceride–Glucose index; TyG-BMI, Triglyceride–Glucose Index plus BMI; HOMA-IR, Homeostasis Model A-Insulin Resistance; HOMA-B, HOMA-β cell; MetS-IR, Metabolic Score for Insulin Resistance; SPISE, the Single Point Insulin Sensitivity Estimator; non-HDL/HDL, non-HDL-cholesterol/HDL-cholesterol ratio; TG/HDL-C ratio, triglycerides/HDL-cholesterol ratio;  MetS-IR, Metabolic Score for Insulin Resistance; DI, Disposition Index; LCI, Lipoprotein Combine Index.

**Table 3 jcm-14-02636-t003:** Prediction of the bladder cancer presence by surrogate markers of insulin sensitivity/resistance.

d.v.	BC (Yes/No)	Odds Ratio	Std. Err.	z	*p* > |z|	95% CI	Pseudo R^2^	n
**i.v.**	**TyG**	**2.10**	**0.67**	**2.34**	**0.02**	**1.13–3.91**	**0.022**	**181**
**i.v**	TyG-BMI	1.01	0.00	1.57	0.12	1.00–1.01	0.01	179
**i.v**	TG/HDL-C ratio	1.21	0.13	1.73	0.08	0.97–1.50	0.017	178
**i.v**	**MetS-IR**	**0.01**	**0.03**	**−2.11**	**0.04**	**0.00–0.74**	**0.018**	**179**
**i.v**	DI	0.54	0.30	−1.10	0.27	0.18–1.61	0.00	156
**i.v**	HOMA-B	1.16	0.11	1.49	0.14	0.96–1.40	0.01	207
**i.v**	HOMA-IR	1.24	0.14	1.88	0.06	0.99–1.55	0.018	156
**i.v**	SPISE	0.84	0.09	−1.59	0.11	0.68–1.04	0.01	176
**i.v**	non HDL/HDLratio	1.22	0.19	1.29	0.20	0.90–1.65	0.00	179
**i.v.**	LCI	1.00	0.00	1.72	0.08	0.99–1.0	0.016	175

Legend: Both TyG and MetS-IR showed significant predictions. Abbreviations: n, number of observations; d.v., dependent variable; i.v., independent variable; BC, Bladder Cancer; BMI, Body Mass Index; TyG, Triglyceride–Glucose index; TyG-BMI, Triglyceride–Glucose Index plus BMI; HOMA-IR, Homeostasis Model Assessment-Insulin Resistance; HOMA-B, HOMA-β cell; MetS-IR, Metabolic Score for Insulin Resistance; SPISE, the Single Point Insulin Sensitivity Estimator; non-HDL/HDL, non-HDL-cholesterol/HDL-cholesterol ratio; TG/HDL-C ratio, triglycerides/HDL-cholesterol ratio; MetS-IR, Metabolic Score for Insulin Resistance; DI, Disposition Index; LCI, Lipoprotein Combine Index.

**Table 4 jcm-14-02636-t004:** Logistic analyses (robust) for the association between triglyceride–glucose index, as well as the metabolic score for insulin resistance and incident bladder cancer grouped by advanced age.

IR Markers	OR	Robust SE	z	*p*	95% CI	Pseudo R^2^	n
TyG > 65 years	1.56	0.70	0.99	0.32	0.65–3.76	0.00	115
**TyG = <65 years**	**2.96**	**1.47**	**2.19**	**0.03**	**1.12–7.83**	**0.06**	**66**
MetS-IR > 65 years	0.08	0.24	−0.85	0.40	0.00–26.49	0.00	113
**MetS-IR = <65 years**	**0.002**	**0.01**	**−2.08**	**0.04**	**0.00–0.69**	**0.05**	**66**

Legend: The robustness was evidenced in older patients when stratified analysis was conducted based on a cut-off of age. Abbreviation: n, number of observations; TyG, Triglyceride–Glucose index; MetS-IR, Metabolic Score for Insulin Resistance.

**Table 5 jcm-14-02636-t005:** The diagnostic reliability of the triglyceride–glucose index.

Patients Classified	True BC	True No BC	Total
Positive	n = 75	n = 52	n = 127
Negative	n = 24	n = 30	n = 54
Total	n = 99	n = 82	n = 181

Legend: Interestingly, the BC presence was correctly diagnosed in 75.7% of the cases, but not its absence. BC, presence of bladder cancer; no BC, absence of bladder cancer; n, number of patients.

## Data Availability

New data were created, while data from records are unavailable due to privacy or ethical restrictions, in order to maintain anonymity.

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
