# Peer review of "Which Surrogate Marker of Insulin Resistance Among Those Proposed in the Literature Better Predicts the Presence of Non-Metastatic Bladder Cancer?"

_jcm, 2025, doi:10.3390/jcm14082636_

Round 1
Reviewer 1 Report
Comments and Suggestions for Authors
Thank you for sending your paper to this journal.
I have to main things to point out.
First, I cannot understand if you only enrolled diabetic patients or you did not have any criterion regarding their diabetic status. If you only enrolled diabetic patients, I suggest you mention this in the title, introduction and material chapters.
Second, the smoking status has to be discussed more seriously, as it is the most important risk factor for bladder cancer. According to the WHO, one can be considered a non smoker 5 years after he/she quit smoking. On the other hand, we don't know if this definition applies to the risk for bladder cancer as well. On the other hand, a former smoker might be a person who decided to quit after the first episode of hematuria, and shortly after was diagnosed with BC. Maybe more stratification is needed here.
I kindly ask you to read and cite the following paper on a similar topic: doi: 10.21873/invivo.13159
When you discuss limits, please don't forget the retrospective nature of your study.
I will gladly review an updated version of your work.
Author Response
REVIEWER 1
C= comment; A= answer
Thank you for sending your paper to this journal. I have to main things to point out.
C= First, I cannot understand if you only enrolled diabetic patients or you did not have any criterion regarding their diabetic status. If you only enrolled diabetic patients, I suggest you mention this in the title, introduction and material chapters.
A= Thank you for this intriguing comment, due to the fact that insulin resistance (IR) is the main aim of the study. We are sorry for not have better specified in the Study Design that the data of the patients were analyzed according their chronological admission to our Urology Departments, independently from their health status, obesity (yes/not), T2DM (yes/not), hypertension (yes/not), after undergoing cystoscopy or TURB and having a BC diagnosis confirmed by histology (yes/not).This choice was performed to avoid a bias of selection and to study the illnesses (BC ) as it appears in the real word in individuals (male and females), elderly (yes/not) with symptoms/findings of urinary tract diseases who were admitted to the Departments.The confirmed BC patients were the case and the non BC patients were the controls, who successively were well-matched for age, gender and BMI to avoid unbalanced presence of co-morbidities that could interfere with the results. Furthermore, some other diseases were excluded in suspicion of interference as detailed in Methods. Anyway, we deleted the confounding term “initial” (as occurring in the beginning) substituting it with “consecutive”, and added “as they were admitted” in the sentence concerning the Study Design.
C=Second, the smoking status has to be discussed more seriously, as it is the most important risk factor for bladder cancer. According to the WHO, one can be considered a non smoker 5 years after he/she quit smoking. On the other hand, we don't know if this definition applies to the risk for bladder cancer as well. On the other hand, a former smoker might be a person who decided to quit after the first episode of hematuria, and shortly after was diagnosed with BC. Maybe more stratification is needed here.
A= Precious comment that addresses an key-aspect that we should accurately point out in the Discussion section. Unfortunately, we were not able to eliminate the systematic difference between the ability of participant groups to accurately recall their smoking habit. But, considering that aspect of an utmost importance we add the former sentence plus the well-written of yours in the Limitations section.
C= I kindly ask you to read and cite the following paper on a similar topic: doi: 10.21873/invivo.13159
A= This piece of research perfectly fits the content of our study, having enrolled ten patients with overactive bladder.
C= When you discuss limits, please don't forget the retrospective nature of your study.
A= We pointed out the retrospective nature of the study in the Limitation sections.
C= I will gladly review an updated version of your work.
A= We trust to have correctly followed your comments, which were found very appropriate and useful to ameliorate the content of our manuscript. Every change was opportunely highlighted in the text.
Reviewer 2 Report
Comments and Suggestions for Authors
Dear authors,
This is undoubtedly a very interesting original study on a very common urological disease, with some significant findings. The sample size is large and the methodology is presented in a detailed way, while the statistical analysis is meticulous and the conclusions are well-supported by the results. The biggest drawback of the study is its retrospective design. Comments for the authors:
1) Have the authors caclulated the sterngth of their study (how many patients are needed in each group so as to report statistically significant findings). A power analysis must be clearly presented in the methods section.
2) Did the authors consider studying the association between Lpa and the presence of bladder cancer?
3) The authors need to explain in their methodology section the logic behind including a control group of patients with other diseases, including women.
4) The limitations of the current study, including its retrospective design must be highlighted in the discussion section.
5) Other studies, associating insulin resistance parameters with presence of bladder cancer must be presented in the discussion section.
6) There are several grammatical and linguistic errors that must be fixed. The number of patients must be presented in this form : (n=...).
Comments on the Quality of English LanguageThere are several linguisticand grammatical errors that must be fixed.
Author Response
REVIEWER 2
C= comment; A= answer
Dear authors,
C= This is undoubtedly a very interesting original study on a very common urological disease, with some significant findings. The sample size is large and the methodology is presented in a detailed way, while the statistical analysis is meticulous and the conclusions are well-supported by the results. The biggest drawback of the study is its retrospective design.
A= The study group is planning to follow-up on this cohort of patients over the next years.
Comments for the authors:
C= Have the authors calculated the strength of their study (how many patients are needed in each group so as to report statistically significant findings). A power analysis must be clearly presented in the methods section.
A= In the Methods, subsection Statistics, it was added the following sentence: As post-hoc exploratory examination, the power analysis to establish the minimum sample size was performed calculating the difference of means and SDs (two-sided) of the main outcome in the two groups (No BC and BC patients). In the Results section were given the results of the calculation, i.e., The power analysis established that the total minimum sample was 238 patients with 119 for each group showing a power of 80% and an alpha value of .05 on the basis of the TyG levels, showing a modestly reduced sample size of the controls group.
C= Did the authors consider studying the association between Lpa and the presence of bladder cancer?
A= LAP, surrogate marker of lipid-based insulin resistance, was not studied due to the fact that the values of the waist circumference were not always registered in the records of the selected patients, thus not allowing applying the formula for its calculation. Interestingly enough, results from a very recent paper confronting TyG and LAP in gastrointestinal cancer was added at the end of the Discussion section. i.e., Furthermore, the LAP and the estimated glucose disposal rate, both recent markers assessing lipid-based IR and insulin sensitivity (59, 60) were not assessed due to the scarcity of reported values of the waist circumference in the analyzed records. Anyway, a recent study showed a surprising overlapping OR of both TyG and LAP in predicting gastrointestinal cancer (61).
C=The authors need to explain in their methodology section the logic behind including a control group of patients with other diseases, including women.
A= This is an interesting comment, of which we are particularly grateful to the reviewer.
According to… Tofthagen C. Threats to validity in retrospective studies. J Adv Pract Oncol. 2012 May;3(3):181-3. PMID: 25031944; PMCID: PMC4093311….The single-group design often employed in retrospective studies limits the researchers’ ability to determine cause and effect. Although it is usually not possible to include a control group in a retrospective study, whenever possible a control group should be included to help establish the cause-and-effect relationship.
Having chosen to assess the sensitivity, specificity and thus the reliability of the selected surrogate markers, a well-matched control group was put together to properly run statistics.
We added in the Methods the following sentence: …..Retrospective designs provide a vehicle for research using existing data but can be riddled with threats to both internal and external validity. To minimize such treads, a control group was set up…….. with related reference.
C= The limitations of the current study, including its retrospective design must be highlighted in the discussion section.
A=In the Limitations section, beyond recall bias about the smoking habit and others, the nature of study was mentioned and it was highlighted the lack of an eventual follow-up.
C= Other studies, associating insulin resistance parameters with presence of bladder cancer must be presented in the discussion section.
A= Unfortunately, pieces of research dealing with the comparison among surrogate markers of insulin resistance (IR) were not found in the literature. Nevertheless, we added the following sentence whose content concerns the results of a very large study, stressing the importance of detecting IR in the Discussion section: At our best knowledge, ours was one of the few studies addressing surrogate markers of IR in BC patients, even though T2DM and excessive body weight, both closely linked to IR, as components of MS, are associated with increased susceptibility of BC in a very large population (59).
C= There are several grammatical and linguistic errors that must be fixed. The number of patients must be presented in this form : (n=…).
A= The number of patients was presented as suggested.The Figure 2 was de novo transferred in the main text, being the lowest part of it, i.e., the high risk threshold, unreadable. English language was polished as much as possible.
Finally, all authors mostly appreciated the appropriateness of your comments/suggestions and were indebted to you.
Round 2
Reviewer 1 Report
Comments and Suggestions for Authors
Thank you for taking into account my remarks !
Reviewer 2 Report
Comments and Suggestions for Authors
No additional comments.